# Soil Aggregate Stability and Organic Carbon Content among Different Forest Types in Temperate Ecosystems in Northeastern China

Yanan Liu [1,†], Xin Sui [2,†], Henian Hua [1], Xu Liu [1], Qiuyang Chang [1], Ruiting Xu [1], Mengsha Li [1,3,*] and Liqiang Mu [1,*]

1 Key Laboratory of Sustainable Forest Ecosystem Management—Ministry of Education, School of Forestry, Northeast Forestry University, Harbin 150040, China; lynm@nefu.edu.cn (Y.L.); 1413085125@nefu.edu.cn (Q.C.)
2 Engineering Research Center of Agricultural Microbiology Technology, Ministry of Education & Heilongjiang Provincial Key Laboratory of Ecological Restoration and Resource Utilization for Cold Region & Key Laboratory of Microbiology, College of Heilongjiang Province & School of Life Sciences, Heilongjiang University, Harbin 150080, China; 2017064@hlju.edu.cn
3 Institute of Nature and Ecology, Heilongjiang Academy of Sciences, Harbin 150040, China
* Correspondence: lms19861004@163.com (M.L.); mlq0417@nefu.edu.cn (L.M.)
† These authors contributed equally to this work.

**Abstract:** Soil aggregates play a crucial role in substance and energy cycles in soil systems. The fixation of soil organic carbon (SOC) is closely tied to the safeguarding mechanisms of soil aggregates. Carbon fixation involves the conversion of atmospheric carbon dioxide into organic molecules by autotrophic organisms. Soil aggregates play a significant role in carbon stabilization, allowing for the physical occlusion of SOC. This study focuses on five forest types, *Betula platyphylla*, *Betula dahurica*, *Quercus mongolica*, *Larix gmelinii*, and mixed forests comprised of *Larix gmelinii* and *Quercus mongolica*, in the Heilongjiang Central Station Black-billed Capercaillie National Nature Reserve, northeast of China. This study investigated the soil aggregate stability (SAS) (water sieving) and aggregate-associated organic carbon (AAOC) at different soil depths in five forest types. Our findings demonstrated that fine macro-aggregates (0.25–2 mm) were the main types of soil aggregates among all the forest types. The SAS gradually decreased with increasing soil depth. Notably, broad-leaved forests exhibited relatively high soil stability. The fine macro-aggregates (0.25–2 mm) had the highest AAOC content, and the AAOC level was highest in the topsoil layer. The SAS and AOCC levels of the *Betula platyphylla* forest and *Betula dahurica* forest were higher than those of other forest types and were significantly affected by the forest type, soil depth, and soil physicochemical properties. Collectively, our findings reveal the key factors influencing aggregate stability and the variations in soil organic carbon content in different forest types. These observations provide a basis for studying the mechanisms of soil aggregate carbon sequestration, as well as for the sustainable development of forest soil carbon sequestration and emission reduction.

**Keywords:** forest type; soil aggregates; stability; organic carbon; soil physical and chemical properties

## 1. Introduction

Soil aggregates are the basic units of soil structure and play a crucial role as the main carriers in soil element cycling and energy flow [1]. Their quantity and distribution can distinctly indicate both the soil's erosion resistance and the soil aggregate stability (SAS) [2]. According to the aggregate size, soil aggregates can be divided into macro-aggregates (>0.25 mm), which include coarse macro-aggregates (>2 mm) and fine macro-aggregates (0.25–2 mm), and micro-aggregates (<0.25 mm). In turn, micro-aggregates include micro-aggregates (0.053–0.25 mm), as well as silt and clay fractions (<0.053 mm). Macro- and micro-aggregates mainly affect the stability of the soil structure and soil fertility and water content, respectively [3]. The mean weight diameter (MWD), geometric mean

diameter (GMD), and fractal dimension (*FD*) of soil aggregates can be used as indicators to evaluate the SAS. The SAS is positively correlated with the MWD and GMD, but negatively correlated with the *FD* [4].

Forest ecosystems represent the largest carbon sink among terrestrial ecosystems. The carbon stocks in forest soil, which have a significant carbon sink capacity, surpass above-ground vegetation carbon stocks [5]. Soil carbon mainly deposits in soil aggregates in forests. However, little is known about the organic carbon fixation mechanism in soil aggregates for forests and the interaction between soil organic carbon and soil aggregates [6]. The fixation of soil organic carbon (SOC) is closely linked to the protective mechanism of soil aggregation because SOC can deposit and accumulate gradually in the soil aggregates. SOC exhibits cohesive properties, and a high SOC content can enhance the formation and SAS of soil aggregates. Substantial research has been conducted on the SAS and organic carbon content in various forest ecosystems, focusing on artificial plantations [7], tree stand age variations [8], revegetation [9], and forest conversions to investigate the relationship between the soil aggregate structure and organic carbon content in forest ecosystems [10]. For instance, studies have shown that during regional forest restoration, tropical forest restoration significantly promotes the accumulation of soil carbon content. [11]. Pen et al. reported that the SAS and SOC both increase with plant succession, contributing to the SAS and the accumulation of aggregate-associated organic carbon (AOCC) [12]. Moreover, under the same climate conditions, different forest types exert varying effects on soil aggregates and their organic carbon characteristics due to their biological features [13]. For instance, a study on three types of subtropical forests in the same region revealed that broad-leaved forests and bamboo forests exhibited a higher SAS and AAOC compared to Chinese fir forests. The forest type influences the distribution of total organic carbon in soil aggregates [14]. Currently, the characteristics of the SAS and organic carbon are not fully understood, such as the relationship between soil aggregates and the SOC for different types of natural forests and the research into soil aggregates and organic carbon in the middle- and high-latitude temperate regions needs to be further deepened. Previous studies have reported that freeze–thaw phenomena occur more frequently in mid- to high-latitude ecosystems, leading to the soil aggregates destabilizing and changing the content of dissolved organic carbon [15,16]. This indicates a potential close relationship between soil aggregates and organic carbon. However, we have limited knowledge about the relationship between organic carbon and soil aggregates in temperate and cold–temperate forest ecosystems during non-freeze–thaw processes. Therefore, exploring the relationships of the changes between the SOC and SAS among diverse forest ecosystems is an urgent matter.

The Heilongjiang Central Station Black-billed Capercaillie National Nature Reserve is a typical temperate forest ecosystem. Five typical forest types were selected in the reserve: birch (*Betula platyphylla*) forests, black birch (*Betula dahurica*) forests, Mongolian oak (*Quercus mongolica*) forests, larch (*Larix gmelinii*) forests, and Mongolian oak and larch (*Quercus mongolica* and *Larix gmelinii*) mixed forests. This study sought to analyze the variations in the SAS and SOC in the five aforementioned forest types. Specifically, our study aimed to examine the impacts of different forest types and soil depths on soil aggregate composition and SOC distribution in this high-altitude forest ecosystem. This research provides valuable insights for gaining a better understanding of sustainable development in terms of carbon sequestration and emission reduction in forest soils.

## 2. Materials and Methods

### 2.1. Site Description

The experimental site was located in the Heilongjiang Central Station Black-billed Capercaillie National Nature Reserve, situated in the southwestern part of Heihe City, Heilongjiang Province (Figure 1). This site falls within the transitional zone between the southwestern foothills of the Lesser Khingan Mountains and the Songnen Plain, covering a total area of 988.6 km$^2$. The region exhibits a temperate continental monsoon climate, characterized by long and severe winters and short and cool summers. The average annual temperature is $-0.5$ °C, with a frost-free period lasting 121 days. The soil freeze period lasts

for up to 270 days, with a snow-cover period of approximately 200 days. The maximum soil freezing depth reaches 250 cm. The annual average rainfall is 476.33 mm, and the average relative humidity is 69.2%. This nature reserve is representative of a typical alpine forest ecosystem, encompassing forests, shrubs, wetlands, and meadows. The forests include coniferous forests, coniferous and broad-leaved mixed forests, and broad-leaved forests, with a high forest coverage of 82.4% [17]. The main plant species in this region include the larch, birch, black birch, Mongolian oak, Amur linden (*Tilia amurensis)*, diamond willow (*Chosenia arbutifolia*), aspen (*Populus davidiana*), bird cherry (*Prunus padus*), goat willow (*Salix raddeana*), and northeast China alder (*Alnus mandshuric*).

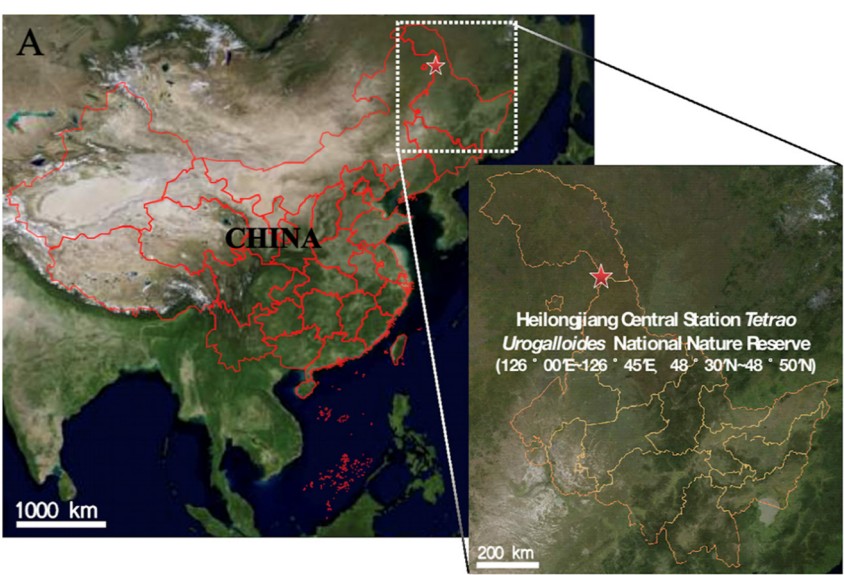

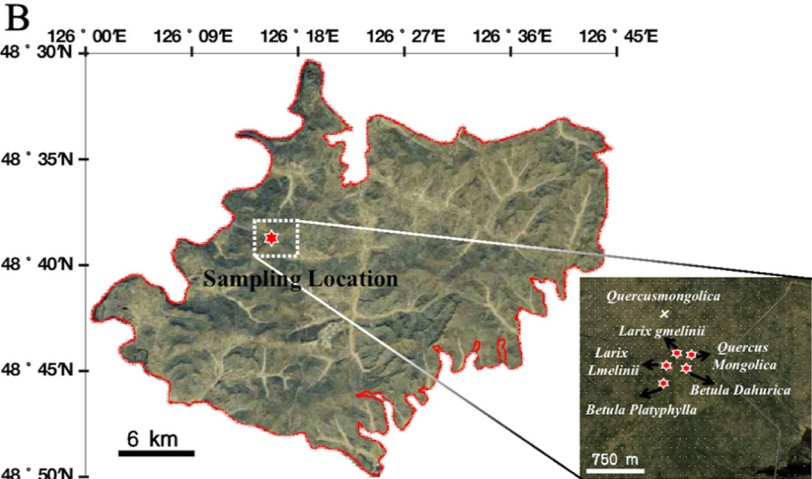

**Figure 1.** Location of sampling sites. (**A**) Map of the sampling site area; (**B**) the specific locations of the sampling sites.

## 2.2. Experimental Design and Soil Sampling

In June 2022, soil samples (0–5 cm, 5–10 cm, and 10–20 cm) were collected using a soil shovel from five typical natural secondary forests (birch, black birch, Mongolian oak, larch, and Mongolian oak and larch mixed forests) in the Heilongjiang Central Station Black-billed Capercaillie National Nature Reserve. The collected soil was identified as mollic epipedon, characterized by its high organic matter content. The topsoil is slightly acidic and is highly fertile due to its elevated nutrient levels. The primary parent materials contributing to its formation are predominantly granite and gneiss. In each of the five forest types, three 10 m × 10 m plots were established in each experimental unit, resulting in a

total of fifteen plots. We collected soil samples from each plot at depths of 0–5 cm, 5–10 cm, and 10–20 cm. One set of soil samples, comprising a total of 15 samples, was mixed in a volume ratio of 0–5 cm:5–10 cm:10–20 cm = 1:1:2. These samples were used to determine the physicochemical properties of the soil within the 0–20 cm depth. Another set of soil samples, totaling 45 samples, was collected based on forest type and soil depth. The soil sample collection process was designed to avoid disturbing the natural structure of the soil, and a soil ring cutter was used to collect undisturbed soil samples for bulk density measurements. The collected soil samples were broken down into smaller clumps based on their natural structure, passed through an 8 mm soil sieve to remove impurities, and then naturally air-dried. Terrain information, such as longitude and latitude, altitude, slope, and slope direction were recorded for each plot. Other forest characteristics, such as stand origin, canopy density, and average breast height diameter, were also measured (Table 1).

**Table 1.** Basic characteristics of the different forest types in the sampling area.

| Forest Type | Latitude and Longitude (E) (N) | Altitude (m) | Origin | Aspect | Slope (°) | Canopy Density (%) | Mean Height (m) |
|---|---|---|---|---|---|---|---|
| *Betula platyphylla* | 125.93019814′ E 50.79863107′ N | 525 | Natural forest | North | 8 | 70 | 17.34 |
| *Betula dahurica* | 125.93453457′ E 50.80034654′ N | 514 | Natural forest | Northeast | 8 | 80 | 9.48 |
| *Quercus mongolica* | 125.93480855′ E 50.80119036′ N | 509 | Natural forest | Northeast | 4 | 70 | 14.85 |
| *Larix gmelinii* | 125.93146803′ E 50.79999506′ N | 519 | Natural forest | North | 10 | 50 | 18.84 |
| *Quercus mongolica* × *Larix gmelinii* | 125.93281439′ E 50.80122973′ N | 508 | Natural forest | North | 5 | 60 | 12.97 |

The soil physicochemical properties were determined using well-established methods. The soil samples were sieved through a 2 mm soil sieve prior to the analysis of soil physicochemical properties (Table 2). Soil pH was measured using a pH meter with a soil-to-water mass ratio of 1:5 (*w/v*). The total nitrogen (TN) concentration was determined using the semi-micro Kjeldahl method [18]. Total phosphorus (TP) was measured using the molybdenum antimony blue colorimetric method, and total potassium (TK) was determined using the sodium-hydroxide-fusion–flame-photometry method [19]. Available nitrogen (AN) was measured using the alkali diffusion method [20]. Available phosphorus (AP) was determined using the sodium bicarbonate extraction method [21]. Available potassium (AK) was measured using the ammonium-acetate-extraction–flame-photometry method [22]. Soil aggregates were measured through water sieving using a soil aggregate analyzer. Dry soil samples weighing 100 g were placed on sieves with apertures of 2.00, 0.25, and 0.053 mm. The soil aggregates in each group were oscillated at a frequency of 60 BPM for 20 min. The collected materials of various particle sizes were dried at 60 °C to a constant weight and then weighed again at room temperature. The SOC content in different soil aggregates was measured using an external heating (oil bath) method [23].

**Table 2.** Soil properties for different forest types within the 0–20 cm soil depth (n = 3).

| Soil Properties | Forest Type | | | | |
|---|---|---|---|---|---|
| | *Betula platyphylla* | *Betula dahurica* | *Quercus mongolica* | *Larix gmelinii* | *Quercus mongolica* × *Larix gmelinii* |
| pH | 4.89 ± 0.12 | 5.34 ± 0.01 | 5.37 ± 0.23 | 5.43 ± 0.08 | 5.50 ± 0.14 |
| BD (g cm$^3$) | 0.77 ± 0.09 | 0.76 ± 0.08 | 0.97 ± 0.28 | 1.09 ± 0.16 | 1.02 ± 0.06 |

**Table 2.** *Cont.*

| Soil Properties | Forest Type | | | | |
| --- | --- | --- | --- | --- | --- |
| | *Betula platyphylla* | *Betula dahurica* | *Quercus mongolica* | *Larix gmelinii* | *Quercus mongolica × Larix gmelinii* |
| TN (mg kg$^{-1}$) | 2056.7 ± 198.0 | 1986.7 ± 268.6 | 1196.7 ± 237.3 | 1673.3 ± 159.0 | 1596.7 ± 217.3 |
| AN (mg kg$^{-1}$) | 75.27 ± 3.71 | 87.1 ± 3.33 | 57.6 ± 4.69 | 85.17 ± 5.09 | 67.07 ± 4.02 |
| TP (mg kg$^{-1}$) | 603.3 ± 82.1 | 433.3 ± 52.4 | 290.0 ± 40.4 | 263.3 ± 18.6 | 276.7 ± 48.1 |
| AP (mg kg$^{-1}$) | 23.52 ± 2.27 | 54.43 ± 2.65 | 18.72 ± 3.31 | 24.28 ± 2.11 | 39.593 ± 3.56 |
| TK (mg kg$^{-1}$) | 8376.7 ± 568.3 | 3383.3 ± 204.3 | 5080.0 ± 555.8 | 6243.3 ± 279.4 | 5313.3 ± 624.4 |
| AK (mg kg$^{-1}$) | 196.1 ± 3.15 | 251.3 ± 17.10 | 190.9 ± 4.65 | 230.2 ± 2.93 | 213.6 ± 4.0 |
| SOC (g kg$^{-1}$) | 54.92 ± 4.87 | 61.44 ± 11.50 | 39.65 ± 5.45 | 36.50 ± 3.39 | 37.35 ± 4.91 |
| SOCS (kg·m$^{-2}$) | 8.55 ± 1.20 | 9.07 ± 1.23 | 7.99 ± 2.33 | 7.86 ± 0.46 | 7.68 ± 1.21 |

Note: TN: total nitrogen; AN: alkali-hydrolyzed nitrogen; TP: total phosphorus; AP: available phosphorus; TK: total potassium; AK: available potassium; SOC: soil organic carbon; SOCS: soil organic carbon stock.

*2.3. Calculations and Statistical Analyses*

(1) Percentage of soil aggregate fractions = mass of the specific aggregate level/total mass of the soil sample × 100%

(2) Mean weight diameter (MWD):

$$\text{MWD} = \sum_{i=0}^{n} m_i \overline{X}_i$$

(3) Geometric mean diameter (GMD):

$$\text{GMD} = \exp\left[\sum_{i=1}^{n} m_i \ln \overline{X}_i\right]$$

(4) Fractal dimension (*FD*):

$$\frac{m_{r<\overline{x}}}{m_0} = \left[\frac{\overline{x}_i}{x_{\max}}\right]^{3-D}$$

*n*: number of groups classified according to aggregate size.

$\overline{X}_i$: average diameter of the aggregate size group.

$X_{max}$: maximum average diameter among all the aggregate size groups

$m_i$: mass of the aggregate in the *i* aggregate size group (g).

$m_0$: total mass of soil aggregates in all aggregate size groups (g).

$m_{r<\overline{x}}$: mass of aggregates smaller than the *i* aggregate size group (g).

These variables are used in calculations related to aggregate size distribution and aggregate stability analysis in soil science.

(5) Aggregate-associated organic carbon contribution rate = (organic carbon content of the aggregate size class (g·kg$^{-1}$))/(total soil organic carbon content (g·kg$^{-1}$)) × 100%.

The differences in soil aggregate size distribution and AAOC content for different soil depths and forest types were tested using one-way analysis of variance (ANOVA), and multiple comparisons were conducted using Fisher's least-significant difference (LSD) test. The effects of forest type, soil depth, and their interaction on SAS and AOCC were analyzed via two-way ANOVA. The effects of soil physicochemical factors on the soil aggregate class, SAS, and AOCC, as well as the correlation between AOCC and corresponding soil aggregate size distribution and SAS, were examined via Spearman correlation analysis. Data processing and plotting were conducted using Microsoft Excel 2019, R (4.2.1) and the SPSS 23.0 software.

## 3. Results

### 3.1. Distribution of Soil Aggregate Sizes

This illustrates the distribution of soil aggregates at different depths for different forest types (Figure 2). The most abundant soil aggregate class among the five types of forests was 0.25–2 mm (37.43%–54.38%). Each size distribution of soil aggregates exhibited a consistent pattern. With increasing soil depth, the proportion of macro-aggregates decreased, whereas the proportion of micro-aggregates gradually increased. The other three soil aggregate size classes (>2 mm; 0.25–2 mm; <0.053 mm) exhibited significant variations ($p < 0.05$), except for the 0.053–0.25 mm size fraction at the 0–5 cm soil depth (Figure 2A). The fraction of coarse macro-aggregates (>2 mm) in the black birch forests was significantly higher than that of the other four forests, but there was no significant difference in the aggregate fractions among the other four forest types. The fraction of fine macro-aggregates (0.25–2 mm) in the birch forest was significantly higher than that in the other four forest types. Additionally, the silt and clay fractions (<0.053 mm) in the larch forests were significantly higher than those in the birch and black birch forests. At the 5–10 cm soil depth (Figure 2B), there were significant differences in the contents of soil aggregates among different forest types ($p < 0.05$). The fractions of the coarse macro-aggregates (>2 mm) and the fine macro-aggregates (0.25–2 mm) were the highest in the black birch and birch forests, respectively. The micro-aggregate (0.053–0.25 mm) fraction, as well as the silt and clay fraction (<0.053 mm), were significantly higher in larch forests compared to black birch forests. At the 10–20 cm soil depth (Figure 2C), the fractions of coarse macro-aggregates (>2 mm) and micro-aggregates (0.053–0.25 mm) were significantly higher in black birch forests compared to those in larch forests and Mongolian oak and larch mixed forests ($p < 0.05$). The content of fine macro-aggregates (0.25–2 mm) was significantly different in the birch forest compared to the other forest types ($p < 0.05$). The silt and clay fraction (<0.053 mm) among the examined forest types was ranked as follows: larch forests > Mongolian oak and larch mixed forests > birch forests > Mongolian oak forests > black birch forests. In particular, larch forests exhibited a silt and clay fraction as high as 38.33%.

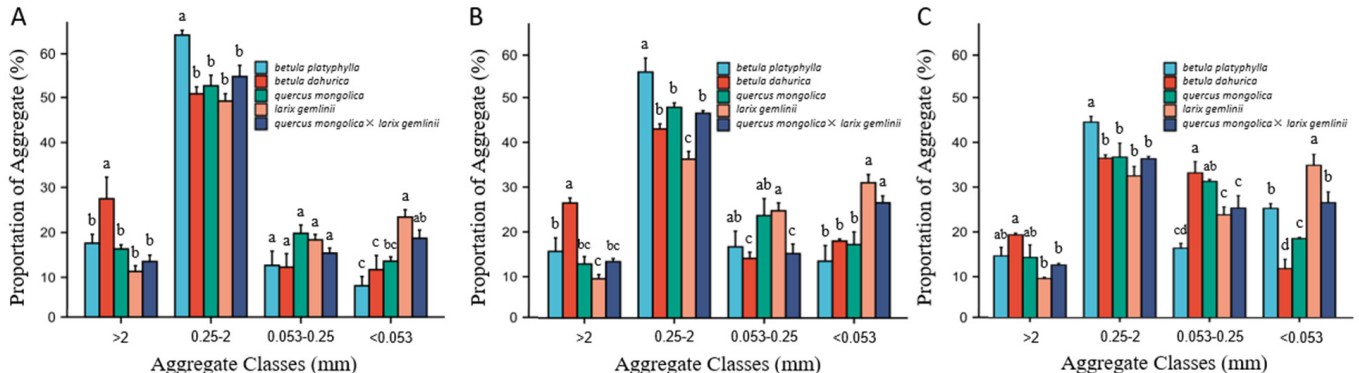

**Figure 2.** The distribution of soil aggregates at different sizes in different forest types: (**A**) 0–5 cm soil depth; (**B**) 5–10 cm soil depth; (**C**) 10–20 cm soil depth. Note: in the same figure, lowercase letters represent comparisons at the 0.05 significance level, indicating significant differences in the content of aggregates with the same aggregate class among different forest types at the same soil depth (a, b, c, d) ($p < 0.05$). The bar represents three replicates of each aggregate class category from each forest type.

### 3.2. Impacts of Different Forest Types on Soil Aggregate Stability

Forests, soil depth, and their interactions have a significant impact on the SAS (Table 3, $p < 0.01$). Table 4 summarizes the results for the SAS for different forest types. The MWD, GWD, and *FD* changed significantly among different forest types and soil depths ($p < 0.05$). As the soil depth increased, the content of soil MWD and GMD tended to decrease, whereas the *FD* exhibited the opposite trend. The content of soil aggregate MWD and GMD were highest in broad-leaved forests and lowest in coniferous forests, whereas the content of soil aggregate *FD* showed the opposite trend, with mixed coniferous and broad-leaved forests

having an intermediate content for the MWD, GMD, and *FD*. Based on the changes in the MWD, GMD, and *FD*, we concluded that the MWD and GMD contents in white birch and black birch forests were significantly higher than those in other forest types, with the MWD value in black birch forests reaching as high as 1.09. This indicates that the soil aggregate structure in birch and black birch forests is good and can better maintain the SAS. The *FD* value in larch forests was significantly higher than that in other forest types, ranging from 2.74 to 2.84.

**Table 3.** The influence of the forest type and soil depth on soil aggregates and aggregate-associated organic carbon.

| Item | Soil Depth | | Forest Type | | Forest Type × Soil Depth | |
|---|---|---|---|---|---|---|
| | *F* | *p* | *F* | *p* | *F* | *p* |
| SAC | <0.001 | 1.000 | <0.001 | 1.000 | <0.001 | 1.000 |
| SAS | 200.121 | <0.001 | 46.064 | <0.001 | 65.534 | <0.001 |
| AAOC | 114.759 | <0.001 | 25.355 | <0.001 | 2.769 | 0.008 |
| AAOCCR | 0.860 | 0.426 | 4.731 | 0.001 | 0.948 | 0.480 |

Note: SAC: soil aggregate class (>2 mm, 0.25–2 mm, 0.053–0.25 mm, <0.053 mm) (180 samples); SAS: soil aggregate stability (MWD, GMD, *FD*) (45 samples); AAOC: aggregate-associated organic carbon (>2 mm, 0.25–2 mm, 0.053–0.25 mm, <0.053 mm) (180 samples); AAOCCR: contribution rate of aggregate-associated organic carbon (180 samples).

**Table 4.** Soil aggregate stability in different forest types.

| Soil Depth/cm | Forest Type | Soil Aggregate Stability Index | | |
|---|---|---|---|---|
| | | MWD | GMD | *FD* |
| 0–5 | *Betula platyphylla* | 1.08 ± 0.05 Aa | 0.75 ± 0.06 Aa | 2.51 ± 0.04 Cc |
| | *Betula dahurica* | 1.09 ± 0.05 Aa | 0.64 ± 0.02 Aab | 2.58 ± 0.03 bCc |
| | *Quercus mongolica* | 0.94 ± 0.03 Ab | 0.52 ± 0.04 Abc | 2.67 ± 0.03 aCb |
| | *Larix gmelinii* | 0.79 ± 0.01 Ac | 0.36 ± 0.01 Ad | 2.74 ± 0.00 Ca |
| | *Quercus mongolica × Larix gmelinii* | 0.90 ± 0.01 Ab | 0.46 ± 0.02 Acd | 2.68 ± 0.01 Cab |
| 5–10 | *Betula platyphylla* | 0.99 ± 0.01 Aa | 0.57 ± 0.02 Ba | 2.63 ± 0.01 Bd |
| | *Betula dahurica* | 1.02 ± 0.05 Aa | 0.52 ± 0.03 Bb | 2.66 ± 0.02 Bc |
| | *Quercus mongolica* | 0.82 ± 0.04 Bb | 0.39 ± 0.01 Bc | 2.73 ± 0.01 Bb |
| | *Larix gmelinii* | 0.65 ± 0.03 Bc | 0.25 ± 0.01 Bd | 2.81 ± 0.03 Ba |
| | *Quercus mongolica × Larix gmelinii* | 0.80 ± 0.02 Bb | 0.33 ± 0.01 Bc | 2.74 ± 0.01 Bb |
| 10–20 | *Betula platyphylla* | 0.84 ± 0.01 Ba | 0.35 ± 0.01 Cb | 2.74 ± 0.01 Ad |
| | *Betula dahurica* | 0.84 ± 0.01 Ba | 0.40 ± 0.02 Ca | 2.77 ± 0.05 Ac |
| | *Quercus mongolica* | 0.74 ± 0.06 Cb | 0.33 ± 0.00 Bbc | 2.80 ± 0.01 Ab |
| | *Larix gmelinii* | 0.59 ± 0.01 Cc | 0.20 ± 0.01 Cc | 2.84 ± 0.01 Aa |
| | *Quercus mongolica × Larix gmelinii* | 0.70 ± 0.02 Cb | 0.29 ± 0.01 Bd | 2.81 ± 0.02 Ab |

Note: the different letters in the same column showed significant difference at the 0.05 level. Different capital letters represent significant differences ($p < 0.05$) in the same forest type but different soil layers with a comparable soil aggregates size distribution (A, B, C); lowercase letters denote significant differences ($p < 0.05$) in the same soil layer but different forest types with a comparable soil aggregates size distribution (a, b, c, d).

### 3.3. AAOC in Different Forest Types

The AAOC is influenced by the synergistic effects of soil depth, forest type, and their interactions (Table 3). Together, forests and soil depth had a significant ($p < 0.01$) effect on the AAOC. This demonstrates the AAOC for five types of forests at different soil depths (Figure 3). At different soil depths, the AAOC tended to increase at first and then decrease with decreasing aggregate class. As the soil depth increases, the organic carbon content in macro-aggregates gradually decreases, while the organic carbon content in micro-aggregates shows an increasing trend. At the 0–5 cm soil depth (Figure 3A), there were significant differences ($p < 0.05$) in the AAOC between different forest types, except for the silt and clay fractions (<0.053 mm), which exhibited no significant differences. The highest AAOC in micro-aggregates (0.053–0.25 mm) was observed in the birch forest,

whereas the other aggregate class fractions exhibited the highest organic carbon content in the black birch forest, reaching 83.34 g·kg$^{-1}$. At the 5–10 cm soil depth (Figure 3B), there were significant differences ($p < 0.05$) in the macro-aggregate (>0.25 mm) AAOC fraction among different forest types, whereas the micro-aggregate (<0.25 mm) fraction exhibited no significant differences. The coarse macro-aggregates (>2 mm) AAOC fraction was significantly higher in the black birch forest compared to Mongolian oak and larch forests. In the fine macro-aggregates (0.25–2 mm). The AAOC in the black birch forest was significantly higher than in the other forest types, except for the birch forest. At the 10–20 cm soil depth (Figure 3C), there were significant differences ($p < 0.05$) in the organic carbon content of the soil aggregates among the different forest types in all aggregate class fractions. The AAOC was highest in the black birch forest for all aggregate class fractions. Except for the silt and clay fractions (<0.053 mm), the AAOC in the black birch forest was significantly higher than that in the larch forest.

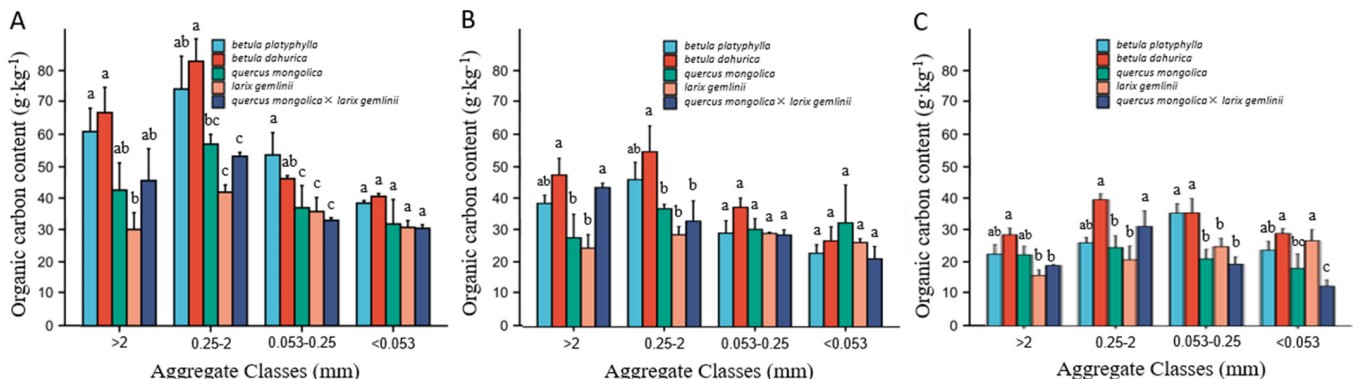

**Figure 3.** Aggregate-associated organic carbon in different forest types: (**A**) 0–5 cm soil depth; (**B**) 5–10 cm soil depth; (**C**) 10–20 cm soil depth. Note: In the same figure, lowercase letters represent comparisons at the 0.05 significance level, and different lowercase letters indicate significant differences among the organic carbon contents of aggregates of the same soil layer, different forest types, and the same aggregate class (a, b, c). ($p < 0.05$). The bars represent three replicates of the organic carbon determination for each aggregate class category from each forest type.

### 3.4. Contribution Rate of the AAOCCR in Different Forest Types

The forest type has a significant impact on the AAOCCR (Table 3). This displays the results for the AAOCCR for different aggregation classes within different forest types (Figure 4). Macro-aggregates were the main contributors to soil organic carbon across the soil profile. However, the contribution of micro aggregates increases with soil depth. The AAOCCR varied among the different forest types, ranging from 4.09% to 70.71%. Macro-aggregates (>2 mm) contributed significantly higher amounts of AOCC in black birch forests compared to other forest types ($p < 0.05$), whereas the highest AOCC levels in fine macro-aggregates (0.25–2 mm) were observed in birch forests, reaching up to 70.71%. The AAOCCR of the micro-aggregates (0.053–0.25 mm) and silt and clay fractions (<0.053 mm) was highest in larch forests and significantly different from other forest types ($p < 0.05$). At the 0–10 cm soil depth, the dominant contributor to the SOC across all forest types was the AOCC of fine macro-aggregates (0.25–2 mm). However, at the 10–20 cm soil depth, the AOCCCR varied among the different forest types. At this point, birch forests, black birch forests, and Mongolian oak and larch mixed forests remained unchanged, whereas the SOC content of larch forests was dominated by the AOCC of micro-aggregates (0.053–0.25 mm) and silt and clay fractions (<0.053 mm). In contrast, Mongolian oak forests were jointly dominated by the AOCC of fine macro-aggregates (0.25–2 mm), microaggregates (0.053–0.25 mm), and silt and clay fractions (<0.053 mm).

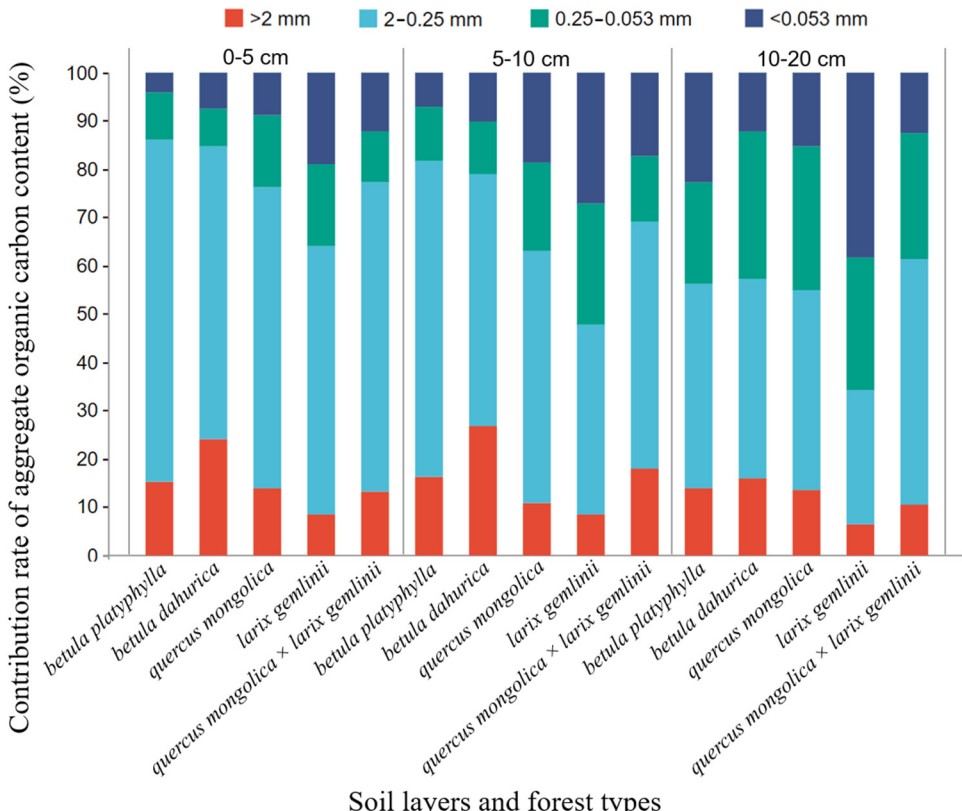

**Figure 4.** Aggregate-associated organic carbon contribution in different forest types.

*3.5. Physicochemical Factors Affecting Soil Aggregate Stability and Aggregate-Associated Organic Carbon*

This illustrates the correlation analyses between soil aggregate characteristics and the physicochemical factors of different forest types in the 0–20 cm soil layer (Figure 5). The values for the soil physicochemical parameters are shown in Table 2. The AAOC was affected by all of the examined soil physicochemical indicators, except for TK. The remaining physicochemical factors (pH, TN, AN, TP, AP, AK, SOC) exhibited a significant positive correlation with macro-aggregates, the AAOC, the MWD, and the GMD, but exhibited a significant negative correlation with micro-aggregates and the *FD*. The SOC is highly positively correlated with macro-aggregates and their AAOC, and the highest correlation coefficient was observed with the fine macro-aggregates (0.25–2 mm) and their AAOC. The TN, AN, TP, AP, and AK all show a significant influence on soil aggregates and their AAOC across different aggregate classes ($p < 0.05$), with varying degrees of impact. Except for pH and TK, all physicochemical factors have a significant influence on the SAS. Among them, the SOC exhibited the highest correlation coefficient with the SAS. Moreover, the SOC was highly positively correlated with the MWD and GMD, and significantly negatively correlated with the *FD* ($p < 0.01$). Soil nutrients can affect the formation of aggregates by changing the accumulation of SOC.

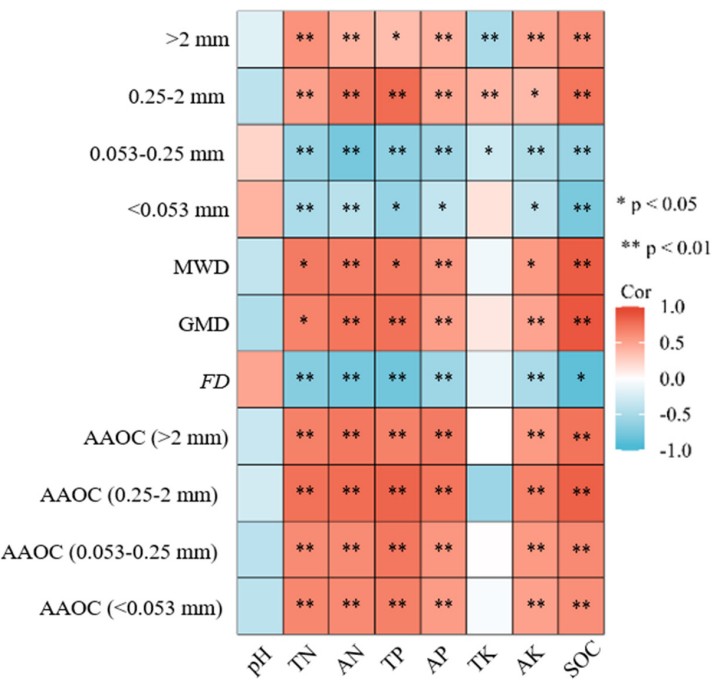

**Figure 5.** The correlation between soil aggregate characteristic values and organic carbon content, with soil physicochemical properties. TN: total nitrogen; AN: alkali-hydrolyzed nitrogen; TP: total phosphorus; AP: available phosphorus; TK: total potassium; AK: available potassium; SOC: soil organic carbon; the asterisks indicate statistically significant differences at a 0.05 and 0.01 level.

### 3.6. Relationships between Soil Aggregate Stability and AAOC

Our findings revealed that there were differences in the correlation between the AAOC and corresponding aggregate class fractions, as well as the SAS, among the five examined forest types (Table 5 and Figure 6). The AAOC levels of the macro-aggregates exhibited a significantly positive correlation with the corresponding size aggregate content ($p < 0.05$). Regarding the AAOC levels of the fine macro-aggregates (0.25–2 mm), except in larch forests, where the correlation was not significant, all other forest types exhibited a highly significant correlation ($p < 0.01$) with the corresponding aggregate class fraction, reaching up to 0.9. The AAOC levels of the micro-aggregates (0.053–0.25 mm) only exhibited a positive correlation with the corresponding aggregate class fractions in Mongolian oak forests, whereas they exhibited a negative correlation in the other forests. The AAOC of the silt and clay fractions (<0.053 mm) was negatively correlated with their corresponding aggregate class fractions in all forest types, with significant correlations observed in Mongolian oak and larch forests ($p < 0.05$). The AAOC was significantly and positively correlated with the MWD and GMD ($p < 0.01$), with correlation coefficients ranging from 0.453 to 0.851. The AAOC was also significant negatively correlated with the *FD*, with the highest correlation coefficient of $-0.828$ observed between the fine macro-aggregate (0.25–2 mm) AAOC and *FD*.

**Table 5.** Correlations between the AAOC levels of different forest types and soil aggregates of corresponding aggregate classes.

| Forest Type | Aggregate Organic Carbon Aggregate Class/mm | | | |
|---|---|---|---|---|
| | 2 | 0.25–2 | 0.053–0.25 | <0.053 |
| *Betula platyphylla* | 0.400 | 0.833 ** | −0.367 | −0.267 |
| *Betula dahurica* | 0.767 * | 0.817 ** | −0.633 | −0.633 |
| *Quercus mongolica* | 0.550 | 0.900 ** | 0.433 | −0.850 ** |
| *Larix gmelinii* | 0.333 | 0.617 | −0.433 | −0.667 * |
| *Quercus mongolica* × *Larix gmelinii* | 0.633 | 0.750* | −0.483 | −0.650 |

Note: * represents $p < 0.05$; ** represents $p < 0.01$.

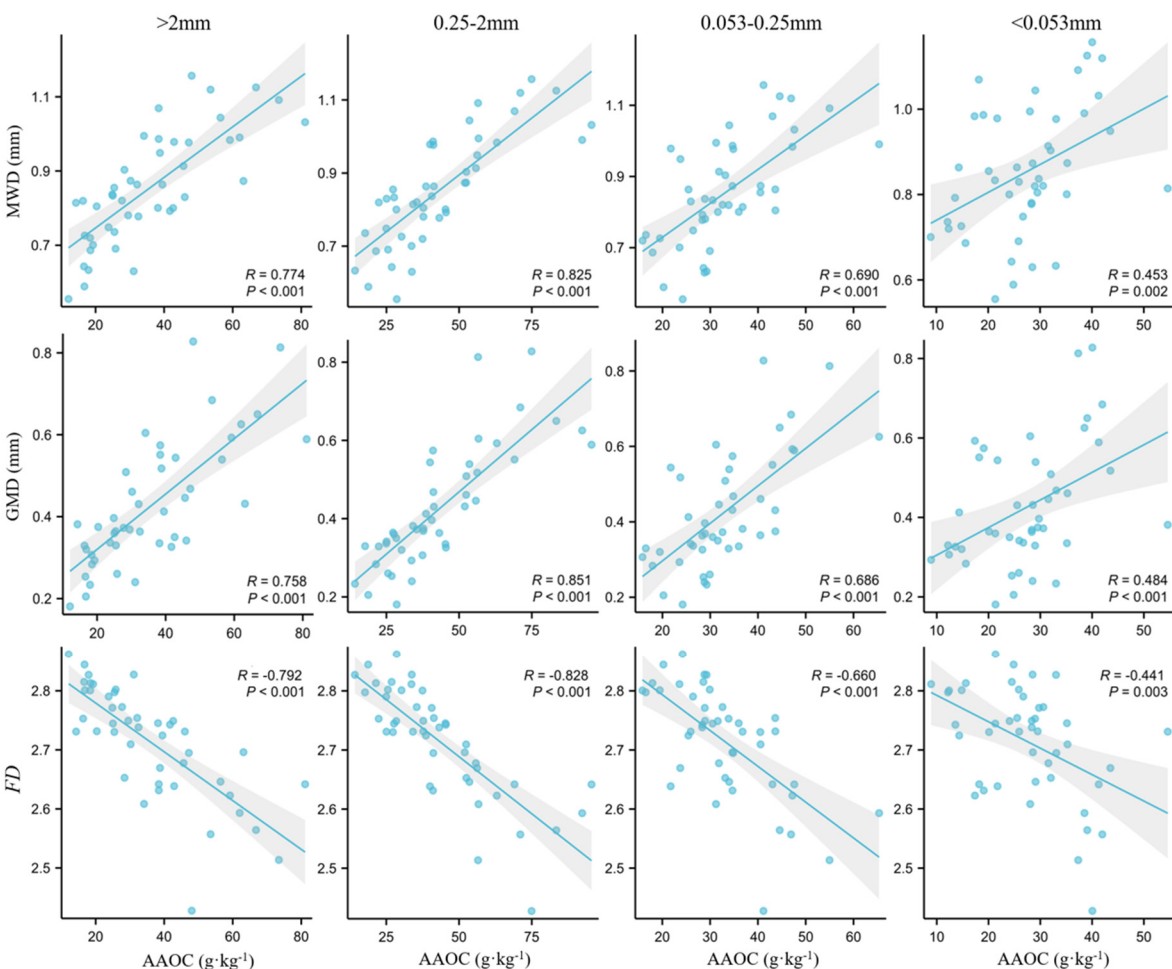

**Figure 6.** Correlations between aggregate-associated organic carbon and aggregated stability indicators.

## 4. Discussion

### 4.1. Changes in Soil Aggregate Distribution and Stability in Different Forest Types

The soil aggregate distribution and stability play an important role in coordinating soil nutrient cycling and nutrient release [24]. The SAS can be affected by the soil nutrient conditions and external factors, and the root distribution and litter decomposition can influence the process of soil aggregation [25]. In this study, both the soil depth and the forest type had significant effects on the soil composition and SAS. The five forest types were mainly composed of fine macro-aggregates (0.25–2 mm), with micro-aggregates accounting for nearly half of the total aggregate content as the soil depth increased. This may be because the soil in the study area froze during the winter due to the low average annual temperature, which caused changes in the soil structure and disrupted the interconnection between soil particles, leading to the splitting of macro-aggregates into micro-aggregates [26]. When the temperature drops below the optimal range for soil microbes, microbial activity is restricted, and microbes' ability to decompose organic matter is weakened, resulting in a reduction in the binding material of soil aggregates. This is consistent with the findings of Zhu et al. and Wang et al. [27,28]. However, Yang et al. conducted a study on soil aggregates in forests consisting of different tree species in subtropical Central Asia and found that coarse macro-aggregates (>2 mm) were dominant throughout the soil profile, with micro-aggregates accounting for only a small proportion [13]. The micro-aggregate content in the 10–20 cm soil layer was higher in our study compared to the data reported by Yang et al. (2022), which may be primarily related to the unique characteristics of different climatic zones. In particular, the climatic conditions in the region studied herein likely provided favorable soil moisture and nutrient conditions. In our study, under the same

soil type conditions, the macro-aggregate content was higher in birch, black birch, and Mongolian oak forests compared to larch and Mongolian oak and larch mixed forests. In particular, larch forests exhibited a silt and clay fraction as high as 38.33%. This difference can be attributed to several factors. Firstly, the high canopy density of the three broad-leaved forests resulted in a thick litter layer; once a layer of leaf litter accumulates to a certain thickness, it can moderate the impact of rainfall erosion, thereby reducing nutrient loss and mitigating harm to soil aggregates. Secondly, different forest types promote the formation of macro-aggregates to varying degrees through interactions between fine roots, fungal hyphae, and the interweaving and compression of roots underground [29,30]. Additionally, the different types of root exudates can lead to variations in soil microbial communities [31], and broad-leaved forests tend to have a higher abundance of rhizosphere microorganisms compared to coniferous forests [32], which can contribute to the formation of macro-aggregates [33].

In terms of the SAS, the content of macro-aggregates in different forest types in this study exhibited the following order: broad-leaved forests > mixed forests > coniferous forests. This suggests that the ability of larch forests to enhance the SAS is lower than that of the other examined forest types. The MWD and GMD were positively associated with macro-aggregates, whereas the *FD* exhibited a negative association. These findings demonstrate that the forest type has a clear influence on the SAS. Additionally, the SAS may also be related to the physical and chemical properties of soils in different ecosystems. For example, our study demonstrated that soil nutrients have certain effects on the soil composition and SAS. However, pH was not significantly correlated with the characteristics of soil aggregates, which may be due to the similar pH values of the soils in the five forest types, all for which were between 4.89 and 5.50 (Table 2). The SOC, TN, AN, TP, AP, and AK were all positively correlated with macro-aggregates, the MWD, and the GMD, whereas they were negatively correlated with micro-aggregates and the *FD*. As a binding organic substance, SOC can bind micro-aggregates in soil into macro-aggregates, thereby improving the SAS [34–36]. N, P, and K are the main indicators of soil fertility. Both the soil nitrogen fertility and soil phosphorus fertility are influenced by the level of organic matter [21,37]. In addition, there is a strong correlation between the stability of soil aggregates and the soil nutrient content. Soil aggregates are the primary sites for storage and transformation of soil nutrients, effectively enveloping soil nutrients and minimizing microbial decomposition. Nutrients can also facilitate the formation of soil aggregates [38].

### 4.2. Changes in Soil Organic Carbon in Different Forest Types

In this study, the AAOC levels of five forest types decreased with increasing soil depth. This trend can be attributed to the natural state of the study area as a forest with minimal human disturbance and abundant understory vegetation. Most fallen branches and leaves accumulate in the topsoil, providing a material basis for microbial activity through decomposition and the formation of a significant amount of humus. This leads to the surface accumulation of SOC. A higher AAOC, SOC, and SOCS were observed in birch and black birch forests (Figure 3, Table 2). This may be due to the fact that compared to coniferous forests and mixed forests, the microbial secretion in broad-leaved forests and the soil beneath them exhibit higher substance secretion and easier litter decomposition, resulting in a higher organic matter content in the topsoil and a greater carbon content in the soil [39]. This indicates that due to their strong carbon sequestration capabilities, birch and black birch species contribute to SOC accumulation, providing more carbon sources for soil aggregates. Our findings demonstrate that there is a highly significant positive correlation between the AAOC level of fine macro-aggregates (0.25–2 mm) and the corresponding class content, indicating that the class composition of soil aggregates has a greater influence on the AAOC. The majority of SOC is concentrated in the fine macro-aggregates (0.25–2 mm), suggesting that these aggregates have the strongest ability to accumulate SOC and serve as the main carrier of soil organic carbon. This may be attributed to the forest ecosystem offering a favorable habitat for soil organisms and microorganisms; their increased activity

range leads to an accelerated decomposition rate, and the presence of a greater abundance of fungal hyphae in macro-aggregates compared to other aggregate classes enhances the concentration of organic carbon within these aggregates during decomposition [40]. The prevailing dominance of fine macro-aggregates (0.25–2 mm) in the studied area indicates their potential superiority in carbon sequestration over micro-aggregates. Additionally, fine macro-aggregates (0.25–2 mm) offer a larger surface area for organic matter attachment, resulting in a higher SOC. Dorji et al. and Wu et al. also reported that macro-aggregates had a higher organic carbon content and stronger carbon sequestration capabilities than micro-aggregates in different forest types [2,41], which is consistent with the results of our study. On the other hand, micro-aggregates form macro-aggregates through the adhesion of organic matter, resulting in a lower SOC compared to macro-aggregates. As the size of soil aggregates decreases, the turnover rate of organic carbon gradually slows down. Larger soil aggregates have a greater capacity for the microbial decomposition of organic carbon, as microorganisms can only access the aggregated structure for carbon decomposition through the secretion of extracellular enzymes and the expenditure of significant energy [42]. Some studies have indicated that the AAOC levels of silt and clay fractions (<0.053 mm) were highest in different forest soils [43]. Mangalassery et al. also suggested that micro-aggregates have more classes and stronger adsorption capabilities and they can more efficiently protect soil organic matter by increasing the inaccessibility to microorganisms and through mineral interaction [44]. Additionally, their aggregate structure is less influenced by external conditions [45]. Micro-aggregates contain a higher mineral content, thereby providing stronger physical protection for the stability of organic carbon [46]. Indeed, different climate and soil conditions can indeed lead to variations in the decomposition characteristics of soil microbial communities [47], thereby affecting the changes in organic carbon content within different aggregate classes. By studying the distribution of SOC among different aggregates classes, further insights can be gained into the protective mechanisms of the AAOC. In this study, the AAOCCR from different forest types exhibited similar patterns to the variation in aggregate classes within the soil profile. At the 0–10 cm soil depth, the AAOC of coarse macro-aggregates (>2 mm) was comparable to that of fine macro-aggregates (0.25–2 mm). However, the AAOCCR of the coarse macro-aggregates (>2 mm) was significantly lower than that of the fine macro-aggregates (0.25–2 mm). The influence of different soil depths on the stability of aggregates and the SOC is likely to depend on the input of carbon. As surface litter enters deeper soil layers, the input of carbon will decrease.

## 5. Conclusions

This study revealed that aggregates and organic carbon in different forest types are predominantly present in the fine macro-aggregates (0.25–2 mm). As the soil depth increases, the SAS and the content of organic carbon gradually decrease. Notably, when it comes to the soil carbon sink capacity of protected areas, broad-leaved forests such as birch, black birch, and Mongolian oak exhibit a higher SAS and SOC compared to mixed forests of Mongolian oak and larch. The physicochemical properties of soil are closely related to soil aggregates and the SOC in forest soils. The SOC, TN, TP, AP, and AK all affect the SAS to varying degrees. Furthermore, the carbon sequestration effects of broad-leaved forests and mixed broad-leaved forests should be given priority if the carbon sequestration capacity of the reserve is to be fully utilized.

**Author Contributions:** The initial idea for this research was conceived by Y.L.; M.L. and L.M. performed the experiments, collected and analyzed the data, and wrote the manuscript; Y.L., R.X., H.H., X.L. and Q.C. performed some of the lab work; and X.S. revised the manuscript. All authors have read and agreed to the published version of the manuscript.

**Funding:** This research was supported by the Open Grant for Key Laboratory of Sustainable Forest Ecosystem Management—Ministry of Education, School of Forestry, Northeast Forestry University

(KFJJ2023YB04) and the Heilongjiang Provincial Ecological Environmental Protection Research Project (HST2022ST008).

**Data Availability Statement:** The data that support the findings of this study are available on request from the corresponding author.

**Acknowledgments:** We are grateful to Zhenzhu Zhao, leader of the Heilongjiang Zhongyangzhan Black-billed Capercaillie Nature Reserve, for allowing us access to the nature reserve, as well as to Fuyuan Chen, Xianda Li and Qiu Guoliang for their assistance in sample processing.

**Conflicts of Interest:** The authors declare no conflict of interest.

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
