# Peer review of "Soil Aggregate Stability and Organic Carbon Content among Different Forest Types in Temperate Ecosystems in Northeastern China"

_forests, doi:10.3390/f15020279_

Round 1

Reviewer 1 Report (New Reviewer)

Comments and Suggestions for Authors

The article is aimed at studying forest soil carbon sequestration. The authors analyze the variations in soil aggregate stability and organic carbon content among five forest types (Betula platyphylla, Betula dahurica, Quercus mongolica, Larix gmelinii, and mixed forests comprised of Larix gmelinii and Quercus mongolica) in temperate ecosystems in northeastern China. The authors also showed the relationship between the physicochemical properties of the soil (total nitrogen, phosphorus and available phosphorus, potassium) and soil aggregates and the fixation of soil organic carbon. The research results contribute to the understanding of the mechanisms of soil aggregate carbon sequestration, as well as for the sustainable development of forest soil carbon sequestration and emission reduction.

Title

The title agrees with the content of the article.

Introduction

The authors should emphasize the relevance, novelty and significance of the research being conducted. The current state of the problem being studied is not fully disclosed. The authors conclude that there is a lack of research on soil aggregates and organic carbon characteristics in temperate regions of middle and high latitudes, but do not provide references.

Line 158 – “their biological features [26]”. The references number is not in order.

Materials and Methods

Methodological approaches are described quite fully, but require minor clarifications. It is not stated how many soil samples were taken for each forest type. One kilogram of soil per forest type? Soil sampling was carried out at several points and a kilogram of soil was taken from each? Or did the authors sample one kilogram of soil from one point or sample it at several points and a kilogram of soil is the mass of an average sample?

I also have a few recommendations:

-       - additionally provide the nomenclature of soils according to the international classification

-       - indicate the contents of elements in the same units of measurement, better in mg kg-1

Results

The results of the study are presented quite clearly and illustrated with five figures and three tables. In my opinion, the notes on some tables and figures regarding the reliability of differences are not very clear. It might be worth writing down what the different lowercase letters mean.

Discussion

The authors have discussed some of their results quite well.

Conclusions

Conclusions follow from the results and are reasonable. In my opinion, the authors should emphasize the practical and theoretical significance of the research.

Author Response

Dear reviewer1,

We sincerely thank you for your kind work on our manuscript, and thank the editor and all reviewers for their valuable feedback that we have used to improve the quality of our manuscript. According to these considerable suggestions, we have made extensive corrections to our previous manuscript. The reviewer comments are laid out below in italicized font and specific concerns have been numbered. Our response is given in normal font and changes/additions to the manuscript are given in the blue text.

If there are further issues to be clarified, please contact us without hesitation.

Best regards,

Liqiang Mu

Key Laboratory of Sustainable Forest Ecosystem Management-Ministry of Education,

Northeast Forestry University,

26 Hexing Road,

150040 Harbin, China

Reviewer: 1,

  1. Introduction

The authors should emphasize the relevance, novelty and significance of the research being conducted. The current state of the problem being studied is not fully disclosed. The authors conclude that there is a lack of research on soil aggregates and organic carbon characteristics in temperate regions of middle and high latitudes, but do not provide references.

Response: Thanks for your suggestion, I reorganized and added the corresponding references. The corresponding parts are modified as "understanding the relationship between soil aggregates and SOC under different types of natural forests, and there is a need for further in-depth research on soil aggregates and organic carbon in the mid- to high-latitude temperate regions" and "Given the context of global change, it is crucial to investigate the interplay between SAS and organic carbon changes ", and the current status, relevance, novelty and importance of this study are clarified. (Lines 72-84)

  1. Line 158 – “their biological features [26]”. The references number is not in order.

Response: We were really sorry for our careless mistakes. Thank you for your reminder. We have made the necessary corrections to the order of the references. (line 68)

  1. Materials and Methods

Methodological approaches are described quite fully, but require minor clarifications. It is not stated how many soil samples were taken for each forest type. One kilogram of soil per forest type? Soil sampling was carried out at several points and a kilogram of soil was taken from each? Or did the authors sample one kilogram of soil from one point or sample it at several points and a kilogram of soil is the mass of an average sample?

Response: Response: Thanks for your suggestion, we have made the necessary changes to the original manuscript. As far as possible, the method of obtaining soil samples is introduced in detail, including the information of sampling site and sampling amount.(line 123-133, line 138-139)

  1. I also have a few recommendations: additionally provide the nomenclature of soils according to the international classification.

Response: According to the International Soil Classification System, we have made the necessary corrections to the soil categories. (line 120-123)

  1. Indicate the contents of elements in the same units of measurement, better in mgkg-1.

Response: Thank you for your suggestion. We have normalized the appropriate units as mg·kg-1 in Table 2 (line 163).

  1. The results of the study are presented quite clearly and illustrated withfive figures and three tables. In my opinion, the notes on some tables and figures regarding the reliability of differences are not very clear. It might be worth writing down what the different lowercase letters mean.

Response: We have provided revised annotations for the different lowercase letters to indicate their meanings. (line 236-239, line 261, line 263, line 293, line 295)

  1. Discussion

The authors have discussed some of their results quite well.

Conclusions

Conclusions follow from the results and are reasonable. In my opinion, the authors should emphasize the practical and theoretical significance of the research.

Response: Thanks for your advice. We have made necessary additions in the conclusion part of the original manuscript, emphasizing the theoretical and practical significance of the research. (Lines 497-494)

Reviewer 2 Report (New Reviewer)

Comments and Suggestions for Authors

Dear Authors! I with interest read your manuscript entitled: “Variations in soil aggregate stability and organic carbon content among different forest types in temperate ecosystems in northeastern China”. The topic of article has actuality/novelty, and fits to Forests journal scope; also it suitable for consideration in other MDPI journals, like: Land, Water, Sustainability, etc. At almost the article is well written, however I have the following comments and suggestions:

L. 292. How many soil samples were collected (in total and from each plot)?

L. 338. “Anti” spectrophotometric method?

LL. 1021-1023. I little disagree with your statement (but for your conditions it could be correct). For example in article (The impact of no-till, conservation, and conventional tillage systems on erosion and soil properties in Lower Austria) the SOC mostly accumulated in clay (<0.002 mm) particles or in fine fractions (silt (0.063–0.002 mm) + clay).

If you have data about soil bulk density you could calculate carbon stocks in different forest types (for 0-30 cm layer), such data also will be interesting. For example please see “Dynamics of biomass and carbon stocks during reforestation on abandoned agricultural lands in Southern Ural region”.

I suggest to add in Reference list some “fresh” literature (dated 2023/2024 year).

Wish good luck in present and future research. 

Author Response

Dear reviewer2,

We sincerely thank you for your kind work on our manuscript, and thank the editor and all reviewers for their valuable feedback that we have used to improve the quality of our manuscript. According to these considerable suggestions, we have made extensive corrections to our previous manuscript. The reviewer comments are laid out below in italicized font and specific concerns have been numbered. Our response is given in normal font and changes/additions to the manuscript are given in the blue text.

If there are further issues to be clarified, please contact us without hesitation.

Best regards,

Liqiang Mu

Key Laboratory of Sustainable Forest Ecosystem Management-Ministry of Education,

Northeast Forestry University,

26 Hexing Road,

150040 Harbin, China

Reviewer: 2,

  1. 292. How many soil samples were collected (in total and from each plot)?

Response: Thank you for your advice. The sentence has been modified according to the reviewer’s suggestion. After correction, In each of the five forest types,Three 10 m × 10 m plots were established in each ex-perimental unit, resulting in a total of 15 plots. Collect soil samples from each plot at depths of 0-5cm, 5-10cm, and 10-20cm. In total, gather 45 soil samples. (line 123-133)

  1. 338. “Anti” spectrophotometric method?

Response: The method in the sentence has been Renewaled to molybdenum antimony blue colorimetric method. In the molybdenum antimony colorimetric method, "Mo-Sb anti spectrophotometric" is a phrase, It's not that "Anti"spectrophotometric method. It is also referred to as the molybdenum antimony blue colorimetric method. The procedure involves converting phosphorus in the soil into soluble orthophosphate through alkaline fusion or acid digestion, and then quantitatively measuring the phosphorus in the solution using the molybdenum blue colorimetric method. The soil sample is fused with sodium hydroxide to convert all phosphorus-containing minerals and organic phosphorus compounds into soluble orthophosphates. The resulting fusion is dissolved in water and diluted sulfuric acid. Under specific conditions, the phosphate ions in the sample solution react with the molybdenum antimony color reagent, generating phosphomolybdenum blue. The intensity of the color is directly proportional to the phosphorus content, which can be quantitatively determined using spectrophotometry. This method is commonly used for soil total phosphorus determination and finds wide application in various soil science research studies. (line 150)

  1. 1021-1023. I little disagree with your statement (but for your conditions it could be correct). For example in article (The impact of no-till, conservation, and conventional tillage systems on erosion and soil properties in Lower Austria) the SOC mostly accumulated in clay (<0.002 mm) particles or in fine fractions (silt (0.063–0.002 mm) + clay).

Response: Thank you for your advice. We have taken the reviewer's suggestions into consideration. We have also added information on the variation of soil organic carbon across different size fractions in forest ecosystems. Our experimental data indicate that, except for larch forests, macroaggregates contribute more significantly to soil organic carbon (OC) in other forest types, and this OC is relatively high within these macroaggregates. We have further explained the reasons behind these results. Additionally, we have incorporated the reviewers' suggestions into the manuscript and included a discussion specifically addressing this aspect. (lines 440-444, lines 450-454; lines 460-463)

  1. If you have data about soil bulk density you could calculate carbon stocks in different forest types (for 0-30 cm layer), such data also will be interesting. For example please see “Dynamics of biomass and carbon stocks during reforestation on abandoned agricultural lands in Southern Ural region”.I suggest to add in Reference list some “fresh” literature (dated 2023/2024 year).

Response: Thank you for your feedback. This will help us a lot. We added soil bulk density data and calculated carbon stocks for each forest type in 0-20cm soil layers. In addition, we have carefully reviewed the references you provided and reorganized the relevant information. We are also interested in estimating carbon stocks in temperate forests in northern China. Therefore, we have supplemented the corresponding contents in the Discussion section. In addition, we have also added the latest research reports to the references. (line 163, lines 426-432, lines 547-549, lines 573-574, lines 621-623)

Reviewer 3 Report (Previous Reviewer 3)

Comments and Suggestions for Authors

Contrary to the authors response in the letter, soil texture has not been added or I can not found it.

Also sampling scheme needs further explanation. According to the material and methods section, this experiment is determined with 15 soil samples, (five samples for each one of the three depths). The samples were air dried and passed to 8 mm mesh. In the figure 6, I can count more than 40 points for each aggregate diameter class, so it must be some analytical repetitions of soil aggregate stability (this information has not been added). Please, indicate the number of analytical repetitions and indicate in figures 2 and 3 the meaning of the bars and the number of repetitions

Have soil physicochemical properties been determined with the 8 mm aggregate  fraction?

Since the physicochemical properties are presented jointly for the 0-20 cm depth, are the values from the three depths mixed to obtain mean value and standard deviation? Please, indicate the number of repetions in table 2 caption.

SAS was determined on four aggregate size fraction but on table 3 it is presented only one statistical analysis for SAS. Were the data of all fractions mixed to obtain statistics? Please indicate the number of samples in table 3

Nitrogen in soil is mainly stored in organic form. Both soil N fertility and soil P fertility are associated to the organic matter level. The increse of soil organic carbon  means the increase of N and P fertility.

Sometimes is employed 53 µm aggregate fraction and sometimes clay and silt fraction. Please use the same criteria.

Author Response

Dear reviewer3,

We sincerely thank you for your kind work on our manuscript, and thank the editor and all reviewers for their valuable feedback that we have used to improve the quality of our manuscript. According to these considerable suggestions, we have made extensive corrections to our previous manuscript. The reviewer comments are laid out below in italicized font and specific concerns have been numbered. Our response is given in normal font and changes/additions to the manuscript are given in the blue text.

If there are further issues to be clarified, please contact us without hesitation.

Best regards,

Liqiang Mu

Key Laboratory of Sustainable Forest Ecosystem Management-Ministry of Education,

Northeast Forestry University,

26 Hexing Road,

150040 Harbin, China

Reviewer 3

  1. Contrary to the authors response in the letter, soil texture has not been added or I can not found it.Also sampling scheme needs further explanation. According to the material and methods section, this experiment is determined with 15 soil samples, (five samples for each one of the three depths). The samples were air dried and passed to 8 mm mesh. In the figure 6, I can count more than 40 points for each aggregate diameter class, so it must be some analytical repetitions of soil aggregate stability (this information has not been added). Please, indicate the number of analytical repetitions and indicate in figures 2 and 3 the meaning of the bars and the number of repetitions

Response: Thank you for your advice. Information about the properties of the soil has been added to the manuscript. The sampling scheme has also been updated in as much detail as possible, clearly describing the sample size and the number of repetitions. We've added notes to the chart, including the meaning of the bar chart and the number of repetitions. (line 120-133, line 238-239, line 296-297)

  1. Have soil physicochemical properties been determined with the 8 mm aggregate fraction?

Response: Soil physicochemical property determination involves sieving the soil through an 8mm sieve to remove impurities, followed by passing it through a 2mm sieve for further analysis. (line 142-145)

  1. Since the physicochemical properties are presented jointly for the 0-20 cm depth, are the values from the three depths mixed to obtain mean value and standard deviation? Please, indicate the number of repetions in table 2 caption.

Response: The three soil layers are mixed in a volume ratio of 1:1:2 and used to determine the physicochemical properties of the soil in the 0- 20cm layer. The number of repetitions has been indicated in the caption of Table 2. (line 127, line 165)

  1. SAS was determined on four aggregate size fraction but on table 3 it is presented only one statistical analysis for SAS. Were the data of all fractions mixed to obtain statistics? Please indicate the number of samples in table 3

Response: In Table 3, SAS will aggregate the data from all the scores to obtain statistical information, totaling 45 samples. At the same time, the number of samples has been indicated in the caption of Figure 3. (line 229-232)

  1. Nitrogen in soil is mainly stored in organic form. Both soil N fertility and soil P fertility are associated to the organic matter level. The increse of soil organic carbon  means the increase of N and P fertility.

Response: Thank you for the reviewer's suggestions. We have incorporated the suggested additions into the manuscript. (line 416-421)

  1. Sometimes is employed ≤ 53 µm aggregate fraction and sometimes clay and silt fraction. Please use the same criteria.

Response: Thank you for the reviewer's suggestions. We have made the necessary revisions to the corresponding section of the paper.(line 274, line 287-288)

This manuscript is a resubmission of an earlier submission. The following is a list of the peer review reports and author responses from that submission.

Round 1

Reviewer 1 Report

Comments and Suggestions for Authors

General comment

The paper “Variations in soil aggregate structure and organic carbon content among different forest types in a temperate forest ecosystems, northeastern of China” focuses on water stability of aggregate of different classes (>2, 2-0.25, 0.25-0.053 and <0.053 mm) and their C content in 5 forest soils.  Five type of vegetations are studied and 0.5, 5-10 and 10-20 cm soil layers have been investigated. The main aim of the manuscript is the investigation of relationships among aggregate stability, C content and forest types. The topic is not novel, as a very large literature exists, and the data presented did not provide new insights into the understanding this complex relationship (aggregate stability-organic C-plant species). In my opinion, it is due to the experimental protocol which, as presented, allowed a mere characterization of aggregates stability and C content in aggregates. The authors will have to provide a theoretical basis for understanding the stability of organic C in the different investigated vegetation (LL. 85-87). However, they do not have data on the stability of organic C. They just provided a quantification of C in the different aggregate classes.

I guess there is a misunderstanding between “C content” (C storage, C distribution) and “C sequestration”. The mere statistical relationships between organic C and aggregates distribution or stability are not sufficient to give indication on organic C stabilization. They may give indications on the distribution of organic C, but this does not mean C stabilization.

Stability of C in soil depends on several mechanisms. Physical protection provided by soil aggregates (i.e., physical occlusion) is only one of these mechanisms. The authors do not have data related to others mechanisms, so they did not know the relevance of physical occlusion in the whole stabilization process of organic C. Furthermore, in physical occlusion of SOM in aggregates, aggregates pores have a relevant role. However, the authors have not data on aggregate pores.

The authors did not provide any information on the type of soils, soil particle size, parent material, slope. These are factors that can greatly affect both aggregate formation and stability, as well as C distribution in the aggregate classes.

The manuscript is poorly organized, and a large of typos are present. Often, the cited references are not pertinent or not available in their English version (see specific comments).

Below some specific comments are reported.

SPECIFIC COMMENTS

TITLE

change "structure" into “stability”

change into "...in a temperate ecosystem" or "....in temperate ecosystems"

ABSTRACT

L: 14: what does it mean? change into "soil aggregates play a crucial role in substance and energy cycles in soil systems"

L. 15: Carbon fixation is the conversion of atmospheric carbon dioxide into organic molecules by autotrophic organisms. Soil aggregates are strongly involved in carbon stabilization allowing physical occlusion of organic C

L. 16: the authors should indicated the method used for measuring aggregate stability. Was it measured by water-sieving? Specify it

L. 19: to what "variables" did the authors refer to? rephase this sentence, it is unclear

L.24: Wha does "there is the most aggregation on the soil surface" mean? did it  mean that the fine macroaggregates from soil surface are the most stable?

LL. 24-26 Please rewrite this sentence checking your English

LL. 27-30: these sentences have a too much broad sense. The relationship between aggregate stability and soil content in forest soil is well-known. What are the new data of this research given new information on aggregate of forest soils and C?

L. 28: aggregate structure is not a synonym of aggregate stability. The authors measured aggregate stability not structure.

INTRODUCTION

L. 37: Here and in the whole manuscript, avoid to use the term "structure" as synonyms of “aggregate” or other term. Structure refers to the spatial arrangement, so it also refers to pores.

L. 37-38: “According to particle size”.  Change into “According to the aggregate size”. Particle size refers to sand, silt and clay.

LL. 45-46: The soil aggregate stability is measured by the MWD, GMD and FD. Rephrase the sentence.

LL. 46-47:"Forest ecosystems.......in terrestrial ecosystems". I guess this sentence is a part of the paragraph below

L. 51: C fixation is the transfer of atmospheric CO2 to organic substances by plant photosynthesis

L.52 "SOC is a cementitious substance of aggregates, ". What does it mean? Check your English

LL. 53-54: "Soil aggregates can reduce the disruption of the original structure of SOC". What does it mean? Rephrase this sentence avoiding any misunderstanding between structure and aggregation

LL. 51-55: Revise your English

L. 57-58: the cited paper [7] by Qiu et al. (2015) focus on agricultural soils. The management of agricultural soils strongly different from forest soils. The authors should cite appropriate and relevant paper related to their research topic

L. 58: the cited paper [9] is not available

L. 58 The cited paper [8] by He et al.,(2022) focus on Pinus massoniana. The authors did not study Pinus massoniana. Why was this paper cited?

LL.60-63: The cited paper [11] by Ma et al (2022) focus on the effects of thinning on soil aggregation in a 36-year-old Larix principis-rupprechtii plantation. No computations were carried out between young and mature forests. The authors should cite appropriate researches

LL. 66-69: reference is missing

LL. 83-87: in my opinion, looking at the experimental protocol, this paper is a characterization of aggregate stability and C content in aggregate in 5 forest soils characterized by a different vegetation covers. How can this experimental protocol provide a theoretical basis for understanding the stability of organic C?

Stability of C in soil depends on several mechanisms. Physical protection provided by soil aggregates (i.e., physical occlusion) is only one of these mechanisms. The authors have no data related to others mechanisms, so they did not known the relevance of physical occlusion in the whole stabilization process of organic C. Furthermore, the authors measured the aggregate stability in water, but aggregate occlusion is related also to aggregate porosity and pore size distribution in the aggregates. The authors have not data on aggregate pores.

Other critical points are present in the experimental protocol, for example the authors did not provide any information on the type of soils. The type of soils, and the pedogenetic processes in general, greatly affect soil properties and soil aggregates properties. So, different results can occur in different soils (see also general comment)

M&M

paragraph 2.2. which type of soil (or types of soils) did the author investigate?

What is the soil texture?

What is the parent material?

What is the slope of each sites?

Table 2 is not cited in the text.  What is the soil depth at which the soil properties in Table 2 refer to?

paragraph 2.3 equations (1), (2) and (3) are not statistical analyses. They are the calculation of stability indexes.

Statistical analyses began at LL. 163.

Why did the authors use three stability indexes. What is the useful information expected by the use or comparison of the results obtained by these three stability indexes? Did the authors expect different information provided by the three indexes? The authors should specify it

LL 154. 161: they are related to aggregate size distribution and not to particle size distribution

L. 163: "the differences in soil aggregates" What did it mean? differences in soil aggregate stability?

L. 165 and L. 167: on soil aggregates? did the authors mean  on soil aggregate stability?

LL. 165-166: The authors have no data on AOCC characteristics, as they only quantify the C in the aggregate and no characterization of organic C has been provided

L. 168 change into "soil aggregate size distribution"

RESULTS

L: 172: the meaning of the title of paragraph 3.1 is unclear. Rephrase checking your English

L- 173: Figure 2 reports the aggregate distribution and not the aggregate content. specify that the aggregate distribution is after wet-sieving

L. 174 change into "the most abundant soil aggregate class...."

L. 175 What is the granular level?

LL. 177-178: In soil science, particle size refer to sand, silt and clay. Please, change "particle size" in "aggregate class"

LL. 221-222: The MWD and GMD give information on aggregate stability being aggregate stability indexes. Please, rephrase this sentence, as the meaning is unclear.

LL. 222-224: MWD is the lowest in larch. so it is consistent to FD indication. GMD has a different behavior. What is the authors' opinion about the different indication provided by these indexes?

LL. 258-278: How was AAOCCR calculated? What is its meaning?

L. 266: Here and in the whole manuscript (Figures and Tables, too), chck the use of "~". Use "-" rather than "~

LL. 279-300: It is unclear to the reader to what depth the soil and aggregate properties refer to. Did the authors determine soil properties (pH, TN, AN, TP, AP, TK, AK) in 0-5, 5-10 and 10-20 cm soil layer? In Figure 5, to what depth did the properties of the aggregate refer to?

LL. 301-322: Why did not the authors consider the soil layer depth?

L 303_ What is SOACC? is it AAOC?

DISCUSSION

L. 324: what did   "soil aggregate structure" in the title of paragraph 4.1 refer to? Soil aggregate distribution?

L. 325: what did   "soil aggregate composition"  refer to? Soil aggregate stability?

L. 326: Gao et al. (2023) (the cited paper [21]), the English version is not available (the link to the English version doesn't work). From the abstract, it seems that Gao et al. (2023) did not study the soil nutirent cycling and release. They studied the distribution of organic C.

The authors should cited available papers, readable by an international scientific publlic, and pertinent papers.

LL.331-337: First of all, Rooney et al. (2022) studied porosity changes induced by freeze-thaw cycles. So, even if Rooney et al. work is very interesting it is not so pertinent here.

Furthermore, I disagree with the authors statement: the authors stated that due to frozen, a shift of macro-aggregates to microaggregates occurred.  if freeze-thaw cycles occurred in the studied forest soils, will the deeper soil layers  be more affect than surface layers? In surface layers freeze-thaw cycles should be more intense.

L. 337-338: for [24] and [25] cited papers, the English version is not available or the link to the English version doesn't work. The authors should cited available papers, readable by an international scientific publlic

LL.338-341: How did Yang et al. support the discussion at LL. 331-336? Are they pertinent?

L. 344: did the authors refer to soil macroaggragate content?

LL. 345-354: Many soil factors can affect microaggregation. The quality of soil organic matter under coniferous, too. Furthermore, the reader did not known if the soil type, the parent material, the particle size, the slope (and thus erosion/sedimentation processes) among these soils are similar or not. Too many infomration are lacking and all the explanation that the authors can give to explain the different aggregate distrituion among the different forest type are not supported by data and thery are too much speculative.

LL. 360-361: MWD, GMD and FD are used as stability indexes. it is well known

LL.361-374: pH and soil nutrients are known to be related to soil organic C content. So, their effect on aggregate stabilization is should be due to their relationship to C. The atuhors should consider the redundance of their correlations.

LL. 381-384: Can any erosion/sedimentation process differently affect the investigated forest soils? The authors should indicated the sites slope in Table 1

LL. 385-386: Microaggregation can be a very efficient process for C stabilization both by physical occlusion (the pores are generally smaller in microaggregate than in macroaggregates, and they can more efficiently protect SOM by increasing inaccessibility to microorganisms) and by mineral interaction (in microaggregate SOM-mineral interaction should be relevant leading to high SOM stability).  This is clearly  visible in Figure 4 for larch, where both in 5-10 and 10-20 cm microaggregate contribute for >50% to OC. The OC in microaggregate is likely more stabilized than in macroaggregates.

The authors should take in mind that C storage ( C content) is not the same that C stabilization. C stabilization is a process that allow to maintain C for a longer time in soil.

LL. 393-397: The authors should cite pertinent paper. [40] focus on agricultural soils.

[2] stated that “The AAOC of the large macroaggregates constituted for 76%–90% of the total AAOC under all LULC types”. Please note, they refer to C content (C storage) and not to C sequestration.

LL: 408-409: It was quite expected. It is well known that C content increased aggregate stability, so higher content of C should correspond to higher amount of macroaggregate stable in water. Larch has a different behavior likely because SOM differed being from coniferous plants.

These data are in agreement with the literature (since 'the 70s a large literature exists on the different effect of plant species on soil aggregate stability).

LL. 413-414: The meaning of this sentence is unclear. rephrase it.

LL. 415-416: I disagree. For larch the microaggregates contribute for more than 50%

LL. 425-427: The authors should include soil type, as the relevance of the different C sequestration mechanisms can be strongly different on the basis of soil type

CONCLUSION

The data did not allow to sketch such type of conclusions. see my general and previous specific comments

FIGURES and TABLES

Table 1: What is the slope of each sites?

Table 2: What is the soil depth at which the soil properties refer to?

Table 3: how were SAPZ and AAOCCR determined? which did aggregate stability index (MWD, GMD of FS) SAS refer to? which did aggregate class AAOC refer to?

Figure 2: What does the title of y-axis mean?" Quality percentage content? x-axis is aggregate classes (mm)

Figure 3 x-axis is aggregate classes (mm)

Figure 6 the x-axis should be AAOC and not SOC

REFERENCES

L. 469-470 Check this reference. The list of authors is lacking

Comments on the Quality of English Language

English very difficul to understand (see specific comments)

Reviewer 2 Report

Comments and Suggestions for Authors

The article needs to be revised.

1. The article has grammatical errors. These are just a few examples, but there are many more.:

Line 46: “FD [4] Forest” àSuggestion: FD [4]. Forest

Line 60: ”For instance, Studies conducted” à Suggestion: For instance, studies conducted

Line 85: “forest ecosystem, This study” à Suggestion: forest ecosystem. This study

Line 94: “988.6km2” à Suggestion: 988.6 km2   (The SI prescribes inserting a space between a number and a unit of measurement units)

Line 114: “The results was in table 1”. Results is a plural word.

Line 116: “the laboratory, The collected” à Suggestion: “the laboratory. The collected”

Line 121: “The soil physicochemical properties was detected” The word properties is plural, therefore the verb should be “were”.

Line 173: “The soil aggregate content at different soil depths under different forest types (Figure 2)” There is no verb in the sentence.

2. Line 111: “Three 10 m × 10 m plots were established in each plot” à Suggestion: It would be appropriate to use two different words instead of writing "plot" twice. For example: experimental unit.

3. Line 114: “The results was in table 1”à Suggestion: The information collected is presented in Table 1.

4. Line 123: Suggestion: Considering that total nitrogen and total kjeldahl nitrogen are not the same, because the total kjeldahl nitrogen is the sum of the organic bounded nitrogen groups and the ammonium-nitrogen, you can write something like that: “The total N (TN) concentration was analyzed using the semi-micro Kjeldahl method [15]. Total Kjeldahl nitrogen is the sum of the organic bounded nitrogen groups and the ammonium-nitrogen”

Unless you used a variation of the Kjeldahl method that includes nitrates.

5. Lines 131-140: It is written “Province). Weigh 100 grams of air-dried soil sample that has passed through an 8 mm sieve. Place the soil sample on a nested set of sieves with aperture sizes of 2.00, 0.25, and 0.053 mm. Immerse the sample in distilled water for 5 minutes, followed by oscillation at a frequency of 60 BPM for 20 minutes. Sequentially collect the soil aggregates from each sieve of the nested set, rinsing 135 them into an aluminum container using a wash bottle. Aggregates with particle sizes 136 smaller than 0.053 mm should be allowed to settle in a bucket for 24 hours. The super-137 natant should then be discarded, and the aggregates should be rinsed into an aluminum container using a wash bottle. Dry the aggregates from each size fraction in the aluminum container to constant weight at 60℃”.

Suggestion: It seems to be written as a laboratory recipe. The wording of the protocol used needs to be improved.

6. Can you check the doi of reference 20?. https://doi.org/10.15889/j.issn.1002-1302.2018.19.08

The DOI address did not lead me to the article and it would be interesting to have access to the organic matter determination method.

7. Line 145 (Table 2). The units are (g/kg) or (g kg-1) but not (g/kg-1).

 8. From line 149 to 162 does not pertain to statistical analysis. Put in a section other than statistical analysis.

9. Lines 179-180: It is written “The coarse “macro-aggregates (>2 mm) in black birch forests was significantly higher than those of other four forests”, and in line 186-187 is written “The coarse macro-aggregates (>2 mm) was significantly higher in black birch forests compared to the other forest types”.  The same information ….

10. How has the AAOCCR (Aggregate-associated organic carbon’s contribution rate) been calculated?

Comments on the Quality of English Language

The quality of English needs to improve.

Reviewer 3 Report

Comments and Suggestions for Authors

The work is interesting and fit to the journal characteristics, but need to be improved before to be considered for publication.

It seem that the work was written by two independent writers and no one have revised it. The references used in the Introduction are different from the references used in the Discussion section.

Below I indicate some specific comments that should be taken into account.

Introduction. Clarify the aims of the work

Material and methods:

Lines 101-104. The common and scientific names of the species are mixed.

Lines 108-111. Indicate the sampling scheme and the number of soil samples taken

Aggregate stability is related to soil texture. Please indicate soil texture in the plots

Lines 129-142. They correspond to the manufacturer instructions

Please, specify how did the authors determine AAOC. Was particulate organic matter determined?

Table 1 . Units of canopy density

Table 2. Indicate the number of samples and the meaning of the data ( mean, standard deviation?)

Equations 1,2 and 3 are aggregate stability índices, not statistical analysis.

Results.

The quality of the figures must be improved.

There is a confusion between aggregates and particles.

Figure 1. what is quality percentage  content?

Figure 6. What does this figure mean? Stability indices integer all the aggregates. The relationship are very similar in all the aggregate fractions.

Comments on the Quality of English Language

Introduction, material and methods and results must be rewritten. Although I´m not a native english, I have detected many spelling errors and in some cases the meaning is not clear. It is worth having the english of the manuscript reviewed by an english speaker.